

# A new application of multiplex PCR combined with membrane biochip assay for rapid detection of 9 common pathogens in sepsis

Yun Li[1],*, LuJie Zhao[1],*, Jingye Wang[2], Peipei Qi[3], Zhongfa Yang[1], Xiangyu Zou[1], Fujun Peng[1] and Shengguang Li[4]

[1] School of Basic Medical Sciences, Weifang Medical University, Weifang, China
[2] Department of Pathology, Weifang Maternal and Child Health Care Hospital, Weifang, China
[3] The Third Department of Neurology Weifang People's Hospital, Weifang, China
[4] Guangzhou Laboratory, Guangzhou, China
* These authors contributed equally to this work.

Corresponding authors
Fujun Peng,
pengfujun@wfmc.edu.cn
Shengguang Li,
li_shengguang@gzlab.ac.cn

## ABSTRACT

Rapid and accurate identification of specific sepsis pathogens is critical for patient treatment and disease control. This study aimed to establish a new application for the rapid identification of common pathogens in patients with suspected sepsis and evaluate its role in clinical application. A multiplex PCR assay was designed to simultaneously amplify specific conserved regions of nine common pathogenic microorganisms in sepsis, including *Acinetobacter baumannii, Escherichia coli, Klebsiella pneumonia, Pseudomonas aeruginosa, Enterococcus faecalis, Staphylococcus aureus, Staphylococcus epidermidis, Streptococcus pneumonia*, and *Candida albicans*. The PCR products were analyzed by a membrane biochip. The analytical sensitivity of the assay was determined at a range of 5–100 copies/reaction for each standard strain, and the detection range was 20–200 cfu/reaction in a series dilution of simulated clinical samples at different concentrations. Out of the 179 clinical samples, the positive rate for pathogens detected by the membrane biochip assay and blood culture method was 20.11% (36/179) and 18.44% (33/179), respectively. However, by comparing the positive rate of the nine common pathogens we detected, the membrane biochip assay tended to be more sensitive than the blood culture method (20.11% *vs* 15.64%). The clinical sensitivity, specificity, positive predictive value (PPV), and negative predictive value (NPV) of the membrane biochip assay were 92.9%, 93.2%, 72.2% and 98.6%, respectively. Generally, this multiplex PCR combined membrane biochip assay can be used to detect major sepsis pathogens, and is useful for early initiation of effective antimicrobial treatment, and is feasible for sepsis pathogens identification in routine clinical practice.

## INTRODUCTION

Sepsis is a syndrome disease caused by an inflammatory immune response to pathogenic infections, and it remains an important cause of death in hospitalized patients (*Singer*

*et al., 2016*). Sepsis is rapidly progressive, and the lack of proper antibiotic treatment in the early initiation of the disease will greatly increase the mortality rate (*Ferrer et al., 2014*; *Grant, Olson & Gerber, 2015*). Although doctors can use empiric antibiotic therapy in a timely manner, to a certain extent, this can lead to the overuse of antibiotics, the increase of drug-resistant bacteria, and the prolongation of hospitalization time, and the increase of costs (*Menezes et al., 2013*). For clinical patients, rapid and accurate identification of sepsis pathogens is the most effective measure to guide antibiotic treatment.

The current standard method for the clinical detection of sepsis pathogens is a blood culture assay. Under sterile conditions, a certain amount of blood from the patient with suspected sepsis is cultured in blood bottles and stored in the culture instrument. The pathogen was identified by drug sensitivity test after the positive alarm (*Demiray, Koroglu & Altindis, 2016*; *Humphries et al., 2021*). However, the blood culture assay has certain limitations, such as a low positive rate. Although studies have shown that increasing the volume of blood culture and number of their bottles can increase the positive rate of blood culture (*Lee et al., 2007*), blood culture assay can only identify less than 40% of patients with sepsis (*Afshari et al., 2012*; *Skvarc et al., 2013*). In addition, blood culture assay is time-consuming and it usually takes 1–3 days to identify the pathogen (*MacBrayne et al., 2021*; *Scheer et al., 2019*). Thus, blood culture assay cannot meet the need for early and rapid clinical diagnosis and treatment. There is an urgent need to develop more rapid and accurate methods in routine clinical practice.

With the development and application of molecular biology technology, direct extraction of nucleic acids of pathogenic microorganisms from patients' blood for amplification had become an effective method for the rapid identification of pathogens (*Espy et al., 2006*; *Peters et al., 2004*). Multiplex polymerase chain reaction (PCR) technology had been applied to the detection of pathogenic microorganisms in many fields due to its high sensitivities and specificities, such as common diarrhea-causing pathogens (*Huang et al., 2018*), bovine respiratory and enteric pathogens (*Thanthrige-Don et al., 2018*), gastroenteritis (*Zhang, Morrison & Tang, 2015*), *etc.* Multiplex real-time quantitative PCR had also been widely used in the detection of pathogens in sepsis (*Horvath et al., 2013*; *Menezes et al., 2013*; *van de Groep et al., 2018*; *Westh et al., 2009*). However, due to the limited number of fluorescent channels in the fluorescence quantitative PCR instrument, the number of pathogen detection in a single reaction was limited, and it was usually necessary to divide the testing samples into several reactions for detection (*van de Groep et al., 2018*; *Westh et al., 2009*). Some researchers had used high-throughput sequencing technology to extract DNA from patients' blood for high-throughput sequencing to identify pathogens. However, high cost and long cycle limited its application in the clinical diagnosis and early management of sepsis (*Grumaz et al., 2016*; *Gyarmati et al., 2015*; *Horiba et al., 2018*; *Ren et al., 2016*).

In our previous study, a multiplex DNA biochip assay was used to detect and type human-infecting influenza viruses. Eighty-one clinical samples, including 66 positive samples with evident seasonal influenza virus infections, were tested, and the clinical sensitivity and specificity were 95.5% and 100% respectively (*Wang et al., 2016*).

**Table 1 Target pathogenic microorganisms.**

| Category | Bacteria name |
|---|---|
| Gram-negative bacteria | *Acinetobacter baumannii* |
| | *Escherichia coli* |
| | *Klebsiella pneumoniae* |
| | *Pseudomonas aeruginosa* |
| Gram-positive bacteria | *Enterococcus faecalis* |
| | *Staphylococcus aureus* |
| | *Staphylococcus epidermidis* |
| | *Streptococcus pneumoniae* |
| Fungus | *Candida albicans* |

In this study, we used multiplex PCR amplification technology combined with the membrane biochip to simultaneously detect nine common pathogens of sepsis in a single reaction. Total nucleic acid was extracted from the peripheral blood of suspected sepsis patients, followed by a multiplex PCR-based biochip assay. Combined with a multiplex PCR system and the automatic hybridization instrument, the whole detection process could be completed within 6 h.

The semi-sealed automated operation process can greatly reduce the possibility of contamination. This membrane-based biochip assay has a series of advantages such as time-saving, labor-saving, high specificity, high sensitivity, and low cost. Furthermore, when combined with the blood culture that is considered as the gold standard in clinical testing, it could provide better clinical diagnosis and treatment. The aim of this study is to establish a new application for the rapid identification of common pathogens in patients with suspected sepsis and evaluate its role in clinical applications.

# MATERIALS AND METHODS

## Primers and probes

All target region sequences for the nine sepsis pathogens were downloaded from NCBI database (Table 1). MEGA6 software was used to align the target gene sequences, and the alignment results were exported to a *fasta* format file. Then, an in-house python script was utilized to process the obtained *fasta* file to generate the degenerate sequences of the target region. Multiplex PCR primers were designed by FastPCR software based on conservative area of pathogens, the size of the amplification product ranges from 90 to 200 bp, and the probe was designed in the context of each amplicon (Table 2). In order to eliminate the nonspecific cross-reactivity with other sepsis pathogens, the designed primers and probes were BLAST against the NCBI nucleotide database. The reverse primer was labeled with biotin at the 5′-end, and the probe was tailed by poly(T)$_{10}$ and amine groups (NH$_2$) at 5′-end to facilitate immobilization on the membrane chip. Primers and probes were synthesized by Shanghai Sangon Biotech.

**Table 2 Summary of oligonucleotide primers and probes for PCR and membrane biochip assay.**

| Target related pathogens | Target region | Primers and probes | Sequences (5′-3′) | Product size (bp) |
|---|---|---|---|---|
| *Acinetobacter baumannii* | blaOXA | Aba-F | GTGGTTGCCTTATGGTGCTCAA | 157 |
| | | Aba-R | Biotin-ACGAAGYACACACTACGGGTGTT | |
| | | Aba-P | NH$_2$-TTTTTTTTTT-CACGAGCAAGATCATTACCATAGCTTTGTTGA | |
| *Escherichia coli* | phoA | Eco-F | TTTGGAGATTATCGTCACKGCAA | 180 |
| | | Eco-R | Biotin-TTTTACTGACGAGGATGCYTCAA | |
| | | Eco-P | NH$_2$-TTTTTTTTTT-GAGCTGCTGCGCGATTACGTAAAGAAGT | |
| *Klebsiella pneumoniae* | khe | Kpn-F | TTATCCAYACTTCYGGATAGCCCTC | 197 |
| | | Kpn-R | Biotin-GCCCGACGATGCYACTTATCC | |
| | | Kpn-P | NH$_2$-TTTTTTTTTT-GCGCKCCRATBGAAAAACGCTCCGGGCTGTC | |
| *Pseudomonas aeruginosa* | algD | Pae-F | TTGCAAAGTGCATGGRTCGAAGAT | 160 |
| | | Pae-R | Biotin-CCAACTTGRTGGCCTTTCCGT | |
| | | Pae-P | NH$_2$-TTTTTTTTTT-CAACTAGTGGCCATTGGCAGGCATT | |
| *Enterococcus faecalis* | gelE | Efa-F | ACTTCRCTACCGACATGTTCCGT | 110 |
| | | Efa-R | Biotin-GTAACTTCTTCRCYAAYTGGTGACC | |
| | | Efa-P | NH$_2$-TTTTTTTTTT-CTTGTTTTTCCATAATTGTTCCATCTGTAGCRTTYACT | |
| *Staphylococcus aureus* | nucA | Sau-F | TTRGTKGATACACCTGAAACAAAGC | 167 |
| | | Sau-R | Biotin-TAAATATACGCTAAGCCACGTCCAT | |
| | | Sau-P | NH$_2$-TTTTTTTTTT-CCTAAAAAAGGTGTAGAGAAATATGGYCCTGAAGC | |
| *Staphylococcus epidermidis* | SE2232 | Sep-F | TATYTCATTACCAGCGACAGGTGAA | 96 |
| | | Sep-R | Biotin-GCCATAAAGATGAGCGCYCCA | |
| | | Sep-P | NH$_2$-TTTTTTTTTT-AACATTGGATRCCAGTGGCTGTTGTACTCWTGTCATT | |
| *Streptococcus pneumoniae* | exoA | Spn-F | CTTTCTGCCAAAGGACCTACAA | 200 |
| | | Spn-R | Biotin-CGRCCTTCCAAGTCCATGGTAGAA | |
| | | Spn-P | NH$_2$-TTTTTTTTTT-AACACKTRGAAATTTTAGAAGAACTCTTCCCAGGC | |
| *Candida albicans* | ITS | Cal-F | GGATCTCTTGGTTCTCGCATCGA | 194 |
| | | Cal-R | Biotin-AGCAAACCCAAGTCGTATTGCTCA | |
| | | Cal-P | NH$_2$-TTTTTTTTTT-CGCAGCGAAATGCGATACGTAATATGAATTGC | |

## Standard pathogen strain

A total of 21 target-related pathogens and 14 target-unrelated pathogens were collected from American Type Culture Collection (ATCC) and China Center of Industrial Culture Collection (CICC) as quality control to test the specificity and sensitivity of our new detection system. The information and source of pathogenic bacteria strains are shown in Table 3.

## Simulated clinical sample preparation

Among the nine common pathogens in sepsis, only one standard pathogen strain was selected for each pathogen microorganism to simulate clinical samples. Serial dilution of standard pathogen strains was used to determine the assay sensitivity. The concentration of each pathogen was determined by the standard plate count method. The diluted pathogen was added to 200 μl peripheral blood from healthy donors to obtain a series of simulated clinical samples with different concentrations ($1 \times 10^4$, $1 \times 10^3$, $5 \times 10^2$, $1 \times 10^2$, $5 \times 10^1$, $1 \times 10^1$ cfu/ml), and the total DNA was extracted by QIAamp DNA Blood Mini Kit (Qiagen, Hilden, Germany) following the instructions.
**Table 3 List of pathogen strains used to evaluate the specificity of the PCR assays.**

| Detection of target related pathogens | Standard strain | Detection of target unrelated pathogens | Standard strain |
|---|---|---|---|
| *Acinetobacter baumannii* | CICC22933, ATCC19606 | *Enterobacter aerogenes* | CICC10293 |
| *Escherichia coli* | CICC10032, CICC10003, CICC10302 | *Enterobacter cloacae* | CICC21539 |
| *Klebsiella pneumoniae* | CICC10870, CICC22914 | *Proteus mirabilis* | CICC21516 |
| *Pseudomonas aeruginosa* | CICC10419, CICC21636, CICC10351 | *Klebsiella oxytoca* | ATCC700324 |
| *Enterococcus faecalis* | CICC10396, CICC21606 | *Pseudomonas maltophilia* | CICC20702 |
| *Staphylococcus aureus* | CICC10384, CICC10001, CICC10201 | *Serratia marcescens* | CICC10355 |
| *Staphylococcus epidermidis* | CICC10294, CICC10436 | *Enterococcus faecium* | CICC21605 |
| *Streptococcus pneumoniae* | ATCC49619, CICC10913 | *Staphylococcus hominis* | CICC23976 |
| *Candida albicans* | ATCC10231, ATCC60193 | *Cryptococcus laurentii* | CICC32956 |
| | | *Cryptococcus neoformans* | ATCC32609 |
| | | *Candida glabrata* | ATCC15126 |
| | | *Candida Krusei* | ATCC14243 |
| | | *Candida parapsilosis* | ATCC22019 |
| | | *Candida tropicalis* | ATCC1369 |

**Note:**
CICC, China Center of Industrial Culture Collection; ATCC, American type culture collection.

## Patients and samples

A total of 179 peripheral blood samples were collected from February 2021 to February 2022 from different patients who were initially admitted with symptoms of suspected sepsis, including fever, hypothermia, and abnormal heart and respiratory rates in Weifang Maternal and Child Health Care Hospital and Weifang People's Hospital. All the clinical blood samples used in this study were obtained with patients' consent. All the research processes were approved by the Human Ethics Committee of Weifang Medical University, Weifang Maternal and Child Health Care Hospital, and Weifang People's Hospital (No. 2022YX-074).

Clinical blood samples were collected with venipuncture from 179 patients with suspected sepsis, including 107 males and 72 females. The average age was 48, the oldest was 92, and the youngest was 15. None of the patients were treated with antibacterial drugs before sample collection. Routine blood cultures were identified using a VITEK-2 Compact system (bioMe´rieux, Marcy l'Etoile, France) and drug susceptibility was tested according to the instruction manual (*Funke & Funke-Kissling, 2004*). Meanwhile, the nucleic acids of blood samples were extracted with blood collection tubes containing EDTA and stored at room temperature for less than 24 h.

## Multiplex PCR system

The DNAs of clinical samples were extracted from 200 μl peripheral blood by the QIAamp DNA Blood Mini Kit. Multiplex PCR was performed using extracted total DNAs as templates using Multiplex PCR MasterMix (UNG) Kit (Cwbio, Beijing, China). A total of nine pairs of primers were mixed and dissolved at a concentration of 10 μM/pairs in deionized water to specifically amplify the conserved areas of nine pathogens. A reaction system of 50 μl was used, including 20 μl of DNA template, 25 μl of 2 × Multiplex PCR

MasterMix (UNG), 1 µl of each 10 µM primers mixture, and 4 µl of ddH$_2$O. The thermal cycle of amplification was set as follows: 95 °C for 10 min; 35 cycles of 95 °C for 30 s, 60 °C for 30 s, 72 °C for 30 s; and 72 °C for 5 min.

## Preparation of membrane biochip

A total of 11 oligonucleotide probes, including nine specific probes, one positive control probe, and one negative control probe, were diluted with NaHCO$_3$ buffer (0.5 M, pH 8.4) to 10 µM, and then 0.18 µl of each probe was dotted on a nylon membrane chip (Biodyne C, Pall Corporation, New York, NY, USA). The pattern of the probes was shown in Fig. 1A. After the probes dotting, the membrane chips were baked at 80 °C for 2 h, and then placed in a hybridization cassette and sealed for use at room temperature.

## Detection of multiplex PCR products by membrane biochip

The main steps were done using our previous methods (*Wang et al., 2016*). Membrane biochip hybridization was performed in an automatic membrane-based biochip instrument (MFS-8, DRAGONLAB, Beijing, China). The processes of hybridization were controlled by the software installed in the computer connected to the automatic device. The main processes were as follows: (1) 10 µl of product from the multiple amplification reaction was first denatured at 99 °C for 5 min, then rapidly ice-cooled for 5 min. After cooling, the PCR product was directly mixed with 500 µl 2 × SSPE/0.1% SDS (20 × SSPE containing 3.0 M NaCl, 0.2 M NaH$_2$PO$_4$, and 0.02 M EDTA, pH 7.4). The hybridization mixtures were piped into 2 ml tubes, and serially placed onto the sample disks in the automatic hybridization instrument. (2) The hybridization reagents were added to the corresponding reagent disk in the automatic hybridization instrument. The biochip cassettes were put into the chip slots. (3) The hybridization task was automatically performed by the hybridization device. The test procedure is as follows: each biochip cassette was pumped into 300 µl of 100 mM NaOH, and incubated at 37 °C for 8 min, then it was washed with 300 µl of 2 × SSPE/0.1% SDS at 60 °C for 5 min. And, 300 µl of the hybridization mixtures (10 µl of multiplex PCR product in 500 µl of 2 × SSPE/0.1% SDS) were pumped into the biochip cassette and incubated at 42 °C for 20 min. After hybridization was complete, the biochip cassette was washed using the washing buffer (2 × SSPE/0.5% SDS) twice. The biochip was labeled with 300 µl of 1:4,000 streptavidin-alkaline phosphatase (ZSGBBIO, Beijing, China) in 2 × SSPE/0.5% SDS at 42 °C for 15 min, and then washed twice using 300 µl of washing buffer (2 × SSPE/0.5% SDS). Then, the membrane biochip was incubated with 300 µl of BCIP/NBT substrate (Amresco Solon, Ohio, USA) at 37 °C for 10 min, and washed with sterilized water after staining. (4) The results of the membrane biochip were analyzed by an optical scanner.

## Determination of assay detection sensitivity

The genomic DNAs of nine strains of pathogenic microorganisms were used to determine the analytical sensitivity (Table 4). Pathogen genomic DNA was serially diluted in

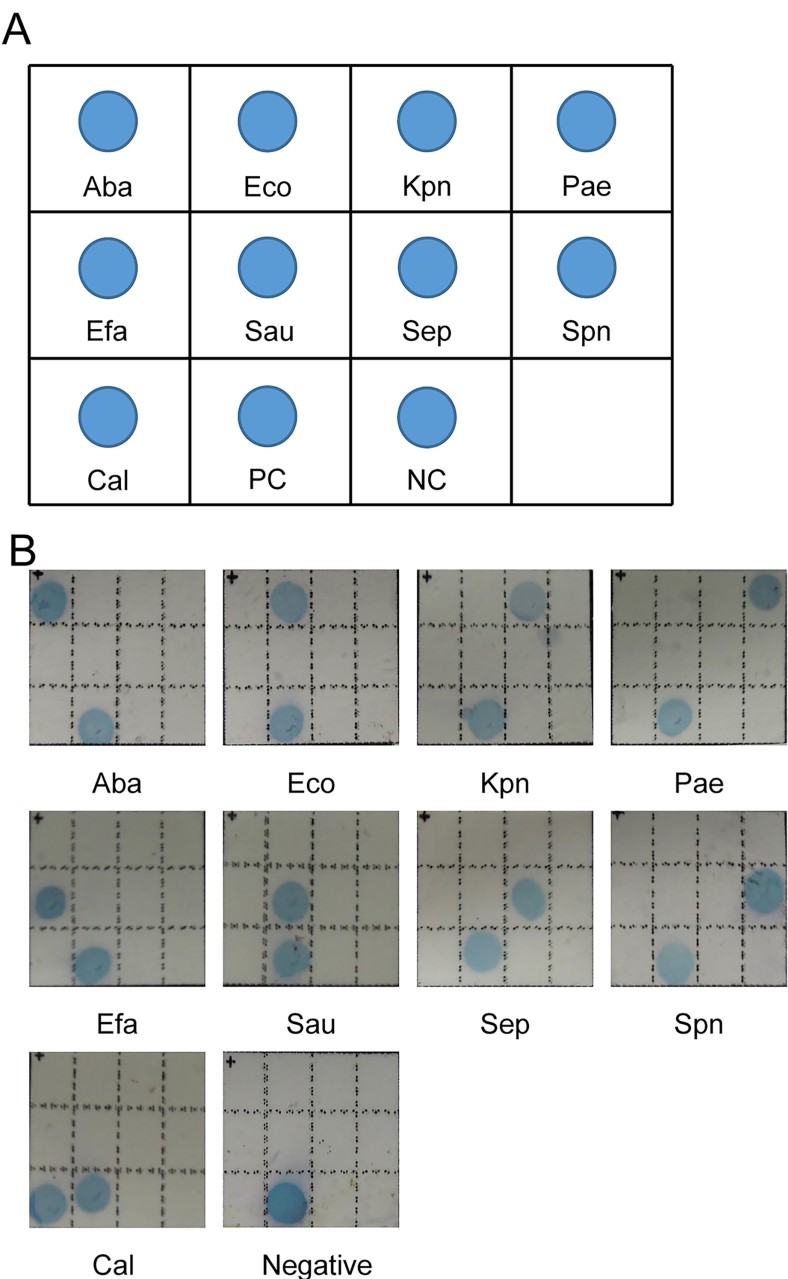

**Figure 1 Representative results of membrane biochip assay.** (A) Scheme of the model of membrane chip. Aba: *Acinetobacter baumannii*, Eco: *Escherichia coli*, Kpn: *Klebsiella pneumonia*, Pae: *Pseudomonas aeruginosa*, Efa: *Enterococcus faecalis*, Sau: *Staphylococcus aureus*, Sep: *Staphylococcus epidermidis*, Spn: *Streptococcus pneumonia*, Cal: *Candida albicans*, PC: positive control, NC: negative control. (B) Pattern of detection of nine pathogenic microorganisms.

nuclease-free water at five different concentrations ($5 \times 10^5$, $5 \times 10^4$, $5 \times 10^3$, $5 \times 10^2$, $2.5 \times 10^2$ copies/ml). The 20 μl of diluted DNAs were used as template for multiplex PCR.

Amplification products were then analyzed in a membrane-based biochip assay with three replicates for each concentration. The detection sensitivity of the assay was defined as the minimum number of DNA copies that were successfully detected in all three replicates.

**Table 4 Detection sensitivity of the membrane-based biochip assay with standard pathogen strains and simulated clinical samples.**

| Pathogens | Standard pathogen strains | | Simulated clinical samples | |
|---|---|---|---|---|
| | Copies ml$^{-1}$ ($\times10^2$) | Copies tube$^{-1}$ ($\times10^0$) | cfu ml$^{-1}$ ($\times10^2$) | cfu tube$^{-1}$ ($\times10^1$) |
| *Acinetobacter baumannii* | 2.5 | 5 | 1 | 2 |
| *Escherichia coli* | 2.5 | 5 | 1 | 2 |
| *Klebsiella pneumoniae* | 50.0 | 100 | 10 | 20 |
| *Pseudomonas aeruginosa* | 2.5 | 5 | 1 | 2 |
| *Enterococcus faecalis* | 2.5 | 5 | 1 | 2 |
| *Staphylococcus aureus* | 5.0 | 10 | 5 | 10 |
| *Staphylococcus epidermidis* | 2.5 | 5 | 1 | 2 |
| *Streptococcus pneumoniae* | 5.0 | 10 | 5 | 10 |
| *Candida albicans* | 2.5 | 5 | 1 | 2 |

## Statistical analyses

Statistical analyses of all data were performed using SPSS 17.0. Pearson $\chi2$ test was used to assess the significance of differences in positive results frequencies between membrane biochip assay and blood culture methods. A two-sided *P*-value < 0.05 was considered statistically significant.

## RESULTS

A total of nine pairs of specific primers were used separately to amplify the extracted microbial genomic DNA of septicemic related pathogens (Fig. 1B and Table 4).
The detection sensitivity of genomic DNA for septicemic pathogens was determined at a range of 5–100 copies/reaction. The sensitivity of this assay was also determined by serial dilution of simulated clinical samples at different concentrations, and the detection range was 20–200 cfu/reaction.

To test our identification system for sepsis pathogens, we collected 179 clinical samples and used multiplex PCR combined with membrane biochip assay to test the performance. Meanwhile, the same clinical samples were also tested by the standard blood culture method. The test results were shown in Tables 5 and S1. A total of 36 positive clinical samples were identified by the membrane biochip method, and 33 positive samples were identified by the blood culture method. There were 26 positive samples with consistent results from both methods, and 10 samples showed positive results only in the membrane biochip system while showing negative results in blood culture. Meanwhile, seven samples showed positive results in blood culture and negative results in the membrane biochip system. Among these seven samples, five pathogenic microorganisms were the target unrelated pathogens that were detected by the blood culture assay but not the membrane biochip system due to the lack of the related primers. Besides, the multiplex PCR productions of 179 samples were detected using agarose gel electrophoresis, and the results were in full agreement with those obtained by membrane-based biochip (Table S1).

**Table 5 Number of sepsis pathogens detected with membrane biochip or blood culture.**

| Pathogens | Membrane biochip | Blood culture | Consistent detection results | Membrane biochip only | Blood culture only |
|---|---|---|---|---|---|
| Detection of target related pathogens | | | | | |
| *Acinetobacter baumannii* | 5 | 4 | 4 | 1 | 0 |
| *Escherichia coli* | 8 | 6 | 5 | 3 | 1 |
| *Klebsiella pneumoniae* | 6 | 5 | 4 | 2 | 1 |
| *Pseudomonas aeruginosa* | 3 | 3 | 3 | 0 | 0 |
| *Enterococcus faecalis* | 3 | 2 | 2 | 1 | 0 |
| *Staphylococcus aureus* | 4 | 3 | 3 | 1 | 0 |
| *Staphylococcus epidermidis* | 4 | 3 | 3 | 1 | 0 |
| *Streptococcus pneumoniae* | 2 | 1 | 1 | 1 | 0 |
| *Candida albicans* | 1 | 1 | 1 | 0 | 0 |
| Subtotal | 36 | 28 | 26 | 10 | 2 |
| Detection of target unrelated pathogens | | | | | |
| *Klebsiella ornithinolytica* | / | 1 | | / | 1 |
| *Stenotrophomonas maltophilia* | / | 1 | | / | 1 |
| *Escherichia hermannii* | / | 1 | | / | 1 |
| *Enterococcus faecium* | / | 1 | | / | 1 |
| *Enterobacter aerogenes* | / | 1 | | / | 1 |
| Subtotal | / | 5 | | / | 5 |
| Total | 36 | 33 | 26 | 10 | 7 |

**Table 6 Comparison between membrane biochip assay and blood culture for all the pathogens and nine target pathogens in 179 suspected sepsis.**

| | No. of membrane biochip assay (%) | No. of blood culture (%) | *P*-value |
|---|---|---|---|
| All the pathogens | | | |
| Positive result | 36 (20.1%) | 33 (18.4%) | 0.688 |
| Negative result | 143 (79.9%) | 146 (81.6%) | |
| Nine target pathogens | | | |
| Positive result | 36 (20.1%) | 28 (15.6%) | 0.270 |
| Negative result | 143 (79.9%) | 151 (84.4%) | |

**Note:**
    *P*-value, Person test.

In all clinical samples, there was no significant difference in the positive rate of sepsis pathogens between the membrane biochip assay and blood culture method (20.11% *vs* 18.44%), as shown in Table 6. However, in our study, the membrane biochip assay could have higher sensitivity than blood culture (20.11% *vs* 15.64%) to detect the nine common pathogens (Table 6).

In order to evaluate the performance of our detection system, we introduced some metrics including sensitivity, specificity, positive predictive value (PPV), and negative

**Table 7 Performance of the membrane biochip assay.**

| Membrane biochip result | Blood culture | | | Sensitivity | Specificity | PPV | NPV |
|---|---|---|---|---|---|---|---|
| | Positive | Negative | Total | | | | |
| Positive | 26 | 10 | 36 | 92.9% | 93.2% | 72.2% | 98.6% |
| Negative | 2 | 136 | 138 | | | | |
| Total | 28 | 146 | 174 | | | | |

**Note:**
PPV, positive predictive value; NPV, negative predictive value.

predictive value (NPV) in 174 clinical peripheral blood samples, excluding five samples caused by the target-unrelated pathogens. The results of the membrane biochip assay were compared with those of the gold-standard blood culture method. The clinical sensitivity, specificity, PPV, and NPV were 92.9%, 93.2%, 72.2%, and 98.6%, respectively (Table 7).

## DISCUSSION

With the development of modern medicine and molecular biology technology, a variety of detection methods for infectious diseases have been established, including microbial culture, serological methods, immunological detection, PCR-based detection, gene chip-based detection and the next generation sequencing technology (*Deng et al., 2021*; *Steele, Orefuwa & Dickmann, 2016*). Especially in recent years, based on the traditional detection methods, molecular diagnosis has been developed towards high-throughput, multi-index, convenience, and rapid detection (*Duan et al., 2021*; *Gu, Miller & Chiu, 2019*). The design concept is to perform multiplex diagnosis in one test. The potential pathogens of the disease were screened in parallel with multiple indicators to overcome the shortcomings of the single-reaction system in the prior art, which were time-consuming and labor-intensive (*MacBrayne et al., 2021*; *Scheer et al., 2019*). These technologies generally integrated a variety of existing technologies, such as multiple nucleic acid amplification, multi-index probe detection, bioinformatics-assisted design technology, *etc*. In this study, we demonstrate a novel application of multiplex PCR combined with membrane biochip assay with a short running time, low reagent consumption, and low cost. This assay also exhibits the advantages of easy operation, safety in use, high throughput, low pollution, *etc*. This is a new product design trend in the field of molecular diagnostics, and it is more suitable for clinical detection applications than traditional methods.

In our research, multiple amplification techniques were combined with membrane biochips to simultaneously detect nine common sepsis pathogens in a single reaction. The sensitivity of the assay varied in genomic DNA of septicemic pathogens and simulated clinical samples, and the detection range was 5–100 copies/reaction and 20–200 cfu/reaction, respectively, which showed similar phenomena with the previous studies (*Thanthrige-Don et al., 2018*; *Wang et al., 2016*). *Wang et al. (2016)* reported that the sensitivity for different viruses varied in the range of 2–100 copies/reaction using the multiplex RT-PCR combined with membrane biochip assay. *Thanthrige-Don et al. (2018)* reported that the analytical sensitivities of multiplex PCR could detect 1–10

copies/reaction in 21 gene targets, even with the lowest sensitivity of 1,000 copies/reaction in the BtoV N gene. The variation in sensitivity may be caused by multiplex factors, including primers and probes specificity, annealing temperature, template characteristics, taq DNA polymerase concentration, *etc*. (*Chanu et al., 2022*; *Markoulatos, Siafakas & Moncany, 2002*). Finally, we evaluated our analysis with a batch of clinical blood samples from Chinese hospitals.

Compared to the previous studies from our laboratory, the clinical sensitivity and specificity in the study had reduced to 92.9% and 93.2%, respectively (*Wang et al., 2016*). The mainly reasons were as follows: (1) There was a difference in the samples' sizes. A total of 174 patients were applied to evaluate the clinical sensitivity and specificity, which may be closer to the actual value than the 81 samples in the study by *Wang et al. (2016)*. (2) There was a difference in the detective pathogens. In this article, nine microorganisms contained cytomembrane, even cytoderm, which increased the difficulty of DNA extraction and decreased the DNA concentrations. However, viruses did not contain the cellular structure. (3) There was a difference in the primers and probes. Both our study and *Wang et al. (2016)* study used different primers and probes, and different annealing temperature to amplify the different target regions, which may lead to the difference in the amplification efficiency. Besides, primers and probes were designed based on the conserved regions in the genomic DNA of the respective pathogenic microorganisms. In order to detect all possible pathogens strains, degenerate primers and probes design strategies were used (*Dugat-Bony et al., 2012*). Although degenerate sequences can target various strains that cover the same pathogenic microorganism (*Hugerth et al., 2014*), the strategy of using degenerate sequences may reduce the detection sensitivity and specificity. (4) There was a difference in DNA concentration. We usually extracted the total DNA from 200 μl of blood. If the blood volume was increased, such as 5 ml, for sample extraction, the detection sensitivity would be greatly improved. Then, we re-extracted the DNA from 5 ml blood for two patients with the negative results using the membrane biochip assay, and performed the above-described processes again. The results showed positive results instead of the previous negative results. Though the clinical sensitivity and specificity were slightly lower than the previous study, the differences did not emerge (Table S2).

Because membrane-based biochip hybridization does not require sophisticated probe printing and signal detection instruments, it is more suitable for low-density DNA chip analysis. Our infectious pathogen detection system for sepsis is simple to operate and has a low cost. Up to eight samples can be tested at one time, and the entire hybridization time is less than 2 h. Furthermore, only one computer is needed to connect and control multiple hybrid instruments, and it could be adjusted as needed. Since the whole hybridization process is conducted in a semi-closed system, contamination can be significantly reduced.

The membrane-based biochip hybridization assay includes DNA extraction from patient blood and biochip-based pathogen identification. The whole processes take less than 6 h. Compared with the 1–3 days test detection cycle for blood culture, this assay can detect the infectious pathogens in sepsis earlier. With the ability to identify pathogens earlier and faster, and to guide clinicians to adopt appropriate antibacterial drugs, the assay

can help doctors better control the disease, reduce the mortality of sepsis and the overuse of antibiotics, and eventually reduce medical costs.

Although our detection method has a series of advantages such as a short detection period, high sensitivity and specificity, and a low cost, it also has certain limitations. Many pathogenic microorganisms could cause sepsis (*Singer et al., 2016*; *Skvarc et al., 2013*). The membrane biochip assay was deficient in clinical sensitivity, specificity, PPV, and NPV, because five clinical samples were infected with pathogens that were not the target of our membrane chip detection scope. We are unable to cover all infectious pathogens of sepsis. So, our detection method could not completely replace the blood culture method, which is the gold standard of the clinical test. However, membrane chip assay can be used as an important supplementary detection method to better serve clinical diagnosis and treatment. This technology can be used for the detection of major sepsis pathogens and can be used to promote early initiation of effective antimicrobial treatment. We have included some of the most common infectious pathogens of sepsis in our assay. In the future, in addition to the nine common pathogens in sepsis, more pathogens that caused sepsis will be introduced into the assay, such as *Klebsiella ornithinolytica*, which will be hopefully replace the blood culture. Meanwhile, we will adjust the detection targets according to the distribution pattern of common infectious pathogens in specific regions and hospitals, and further optimize the degenerate primers and probes sequences to improve the sensitivity and specificity for meeting clinical needs.

## CONCLUSION

In this study, we developed a new detection system based on multiplex PCR and membrane chip that allows simultaneous detection of nine common sepsis pathogens with high sensitivity and specificity in one simple operation. This detection system significantly reduces the identification time for pathogens in clinical septic infections at a low cost. It has great application value in the diagnosis of sepsis, which is beneficial for the early initiation of effective antimicrobial treatment. Multiplex PCR combined membrane biochip assay is feasible for pathogens identification of sepsis in routine clinical practice.

## ABBREVIATIONS

| | |
|---|---|
| **ATCC** | American type culture collection |
| **CICC** | China Center of Industrial Culture Collection |
| **NH$_2$** | Amine groups |
| **NPV** | Negative predictive value |
| **PCR** | Polymerase chain reaction |
| **PPV** | Positive predictive value |

## ACKNOWLEDGEMENTS

We thank all the laboratory members, especially Dr. Yongqiang Wang and Professor Taijiao Jiang for their positive suggestions.

### Funding

This work was supported by the Youth Natural Science Foundation of Shandong Province (ZR2021QH367), the Tai-Shan Scholar Program from Shandong Province (No. tsqn202103116); and the Student Innovation and Entrepreneurship Training Program of Shandong Province (No. x2021003). The funders had no role in study design, data collection and analysis, decision to publish, or preparation of the manuscript.

### Grant Disclosures

The following grant information was disclosed by the authors:
Youth Natural Science Foundation of Shandong Province: ZR2021QH367.
Tai-Shan Scholar Program from Shandong Province: tsqn202103116.
Student Innovation and Entrepreneurship Training Program of Shandong ProvinceL: x2021003.

### Competing Interests

The authors declare that they have no competing interests.

### Author Contributions

- Yun Li conceived and designed the experiments, performed the experiments, prepared figures and/or tables, authored or reviewed drafts of the article, and approved the final draft.
- LuJie Zhao conceived and designed the experiments, performed the experiments, prepared figures and/or tables, authored or reviewed drafts of the article, and approved the final draft.
- Jingye Wang performed the experiments, prepared figures and/or tables, authored or reviewed drafts of the article, and approved the final draft.
- Peipei Qi performed the experiments, prepared figures and/or tables, authored or reviewed drafts of the article, and approved the final draft.
- Zhongfa Yang analyzed the data, authored or reviewed drafts of the article, and approved the final draft.
- Xiangyu Zou analyzed the data, authored or reviewed drafts of the article, and approved the final draft.
- Fujun Peng conceived and designed the experiments, authored or reviewed drafts of the article, and approved the final draft.
- Shengguang Li conceived and designed the experiments, authored or reviewed drafts of the article, and approved the final draft.

### Data Availability

The raw measurements are available in the Supplemental Files.

## Supplemental Information

Supplemental information for this article can be found online at http://dx.doi.org/10.7717/peerj.15325#supplemental-information.

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
