# Peer review of "A new application of multiplex PCR combined with membrane biochip assay for rapid detection of 9 common pathogens in sepsis"

_PeerJ, doi:10.7717/peerj.15325_

## Round 0.1 · original submission · Major Revisions

Please attend to the comments and resubmit at the earliest.

Reviewer 1 ·

Basic reporting

The manuscript ‘A new application of multiplex PCR combined with membrane biochip assay for rapid detection of 9 common pathogens in sepsis’ is an interesting study that discusses the development of a new multiplex PCR method combined with membrane biochip assay to detect pathogens in sepsis patients. The manuscript needs major revision before considering for publication in the current journal.
1. The manuscript needs thorough proofreading by a professional English speaker as the manuscript has spelling errors, grammar mistakes, incorrect prepositions and missing articles at many places. This makes it difficult to understand the manuscript. Alignment and spacing also needs to be checked. The manuscript is not in a scientific language and needs improvement as per my opinion.
2. As per the Journal format, funding is not supposed to be in the acknowledgement. Cross-check the references. The references provided are not according to the journal format.

Experimental design

The research article falls within the aims and scope of the journal. Research questions are defined well and are relevant. Methods are described clearly and a thorough investigation has been done.

Validity of the findings

3. Why is the clinical sensitivity and specificity been reduced to 78.8 % and 93.2 % in the current study when compared to the previous study?
4. Discuss the factors that contribute to the low specificity and sensitivity of this assay?
5. Explain how the multiplex PCR works in combination with the chip-based assay in a more detailed way.
6. The results of table 4 are not discussed well enough in the manuscript.
7. What do the consistent results in table 5 signify?
8. Discuss the future directions of the research at the end of the manuscript.
9. Consider including the multiplex PCR results in the supplementary material

Additional comments

10. Line 170: Sentence incomplete
11. Line 172 and 227: Sentence incorrect. Rephrase the sentences.
12. Line 285: Strains
13. The scientific names are not italicized at certain instances. For example, see line no 439

Reviewer 2 ·

Basic reporting

no comment

Experimental design

no comment

Validity of the findings

no comment

Additional comments

The authors address an essential topic that is of interest to many. To devise a fast and reliable method
for the common pathogens in sepsis detection multiplex PCR combined with DNA chip tools were
applied. This approach is considered as innovative, sensitive, selective, low cost, and fast techniques.
The article includes sufficient introduction and background materials, and relevant literature is
appropriately referenced. The structure of the manuscript is very well and clearly organized in accord
with the Journal’s instructions. The experiments and data are appropriate. Methods are described with
sufficient information for reproduction by another researcher. All appropriate raw data are available.
The figure and the tables clearly represent the results.

The proposed technology is very flexible, and once developed it can be applied to other not less
important medical and even ecological problems.

Overall, I have no major issues with the article but there are several minor items that need addressing.
In section 2.5 in the first sentence “The DNA of clinical samples were extracted from 200 microliter
peripheral blood by QIAamp DNA Blood Mini Kit. Multiplex PCR was carried out using extracted total
DNA as template”, 200 microliters should be written as 200 µl

In section 2.5 the sentence “The potential PCR product contamination.” (line 9), does not have any
meaning.

In section 2.8 (line 4) again 20 microliters should be written as 20 µl.

Annotated reviews are not available for download in order to protect the identity of reviewers who chose to remain anonymous.

---

## Round 0.2 · Minor Revisions

Please see reviewer's comments and revise and resubmit at the earliest

Reviewer 2 ·

Basic reporting

no comments

Experimental design

no comments

Validity of the findings

no comments

Additional comments

I accept the re-reviewed version.

Reviewer 3 ·

Basic reporting

This is a revised article, which has significantly improved compared to its original version, and reviewer queries have been adequately addressed. This manuscript presents a novel approach to the rapid identification of bacteria in septic patients, which is of high clinical relevance. The background introduces the reader adequately to the topic and existing research gaps, the methodology is rigorous, the results are adequately presented, and raw data is being shared.
I have a few questions for clarification. In addition, although the English language has significantly improved compared to the original manuscript version, the manuscript still contains some spelling errors, unusual wording, and sentence structures. Some examples are listed below:

* "detective pathogens" in the abstract
* "abuse of antibiotics" in the introduction first paragraph; perhaps "overuse" might be better
* "suspected patient" would better read as "patient with suspected sepsis" (introduction 2nd paragraph)
* "a urgent need" (introduction 2nd paragraph)
* "It can be combined with clinical testing gold standard blood culture to better serve clinical diagnosis and treatment." Please change this phrasing in the introduction last paragraph.
* "form American type culture collection (ATCC)". Please correct to "from American Type Culture Collection" in methods second section.
* "The concentration of each pathogen was detected by the standard plate count method" (method 3rd section). Please change to "determined".
* "obvious symptoms of suspected sepsis". Please delete obvious in methods 4th section.
* "to specificity amplify conserved areas of 9 pathogens" (methods 5th section". Do the authors mean "specifically" instead?
* "In order to make the hybridization device automatically complete hybridization, the detection workflow was set as follow: each biochip cassette was pumped into 300 µl of 100 mM NaOH and incubated at 37℃ for 8 min, and 300 µl of 2 × SSPE/0.1% SDS was used to wash the biochip at 60℃ for 5 min." (methods 7th section). Please rephrase, as the current wording is unclear.
* "non-nuclease water" please change to "nuclease-free water" in methods 8th section.
* "In this study, we demonstrate a novel application of multiplex PCR combined with membrane biochip assay with short a short running time, low reagent consumption, and low cost. This assay also exhibits the advantages of mildness, safety in use, high throughput, low pollution, etc." (discussion first paragraph). Please delete "short". What does "mildness" mean in this context?
* "sepsis infection" (discussion 5th paragraph), please delete "infection". This term has been used several times throughout the manuscript.
* "...and can be used to help early initiate effective antimicrobial treatment" (discussion last paragraph). Please rephrase, for example to "and can be used to promote early initiation of effective antimicrobial treatment".
* "Completing Interests", please change to "Competing Interests".
* "Table S2 Comparison of clinical sensitivity and specificity in different papers". The authors may consider changing the title to "Table S2 Comparison of clinical sensitivity and specificity in different articles".

* Could the authors please add references to the following statement in the discussion first paragraph: "The potential pathogens of the disease were screened in parallel with multiple indicators to overcome the shortcomings of the single-reaction system in the prior art...".
* The first paragraph in the results section seems to be a repetition of parts of the method section and may be deleted.

Experimental design

The research question is well designed and the investigation rigorous. I do have a few remaining questions and/or comments.

* Could the authors please clarify the duration of the experiment: it is 2 hours or 6 hours?

* "Thanthrige-Don et al. reported that the analytical sensitivities of multiplex PCR and subsequent microarray detection could detect 1-10 copies/reaction in 21 gene targets, even with the lowest sensitivity of 1000 copies/reaction in BtoV N gene (Thanthrige-Don et al. 2018)" (discussion 2nd paragraph). I do not fully understand this statement. Could the author please explain what is meant here?

* Were the unrelated bacterial strains that are listed in table 3 also tested using this assay, and if yes what were the findings?

* The last paragraph of the results states "In order to evaluate the performance of our detection system, we introduced some metrics including sensitivity, specificity, positive predictive value (PPV), and negative predictive value (NPV) in 174 clinical peripheral blood samples, excluding 5 samples caused by the target-unrelated pathogens." However, the last paragraph of the discussion states "The membrane biochip assay was deficient in clinical sensitivity, specificity, PPV, and NPV, because 5 clinical samples were infected with pathogens that were not the target of our membrane chip detection scope." Could the authors please clarify this discrepancy, or alternatively present the results with and without the off-target pathogens?

* In the response letter to reviewer 1, the authors state that increasing the blood volume to 5 ml instead of 500 ul improved the detection of pathogens. This was however only tested on 2 samples, why? Perhaps the authors could recommend a larger blood volume for their assay.

Validity of the findings

*I concur with reviewer 1 comment 9, that the authors may consider including the multiplex PCR results in the supplementary material. Otherwise, all data is being shared.

Additional comments

No additional comments

---

## Round 0.3 · accepted · Accept

Accepted for publication as all reviewer comments are answered to satisfaction.